# HUMANEXPERT: UNIFIED MULTIMODAL HUMANOID GENERATION

## ABSTRACT

Though recent advances in unified multimodal understanding and generation have unfolded, building multimodal humanoid agents capable of mimicking core human abilities, such as language understanding, speech, and behavior generation, remains challenging. Symbolic modalities like language rely on discrete tokens, while perceptual modalities such as vision and behavior benefit from continuous representations, making unified understanding and generation across such diverse modalities difficult. Insightfully, by decoupling model parameters across modalities and adopting a modality-expert training strategy, we avoid degrading the original language model's intelligence while enabling the interleaving of continuous and discrete tokens within a unified generative framework. Inspired by this, we propose HumanExpert, a unified multimodal generative model for humanoid agent tasks, synthesizing language, speech, and behavior in one interleaved autoregressive-diffusion framework with a behavior expert. Specifically, HumanExpert employs a mixture-of-experts (MoE) architecture with a modality-independent backbone, where the behavior expert enables human behavior modeling while preserving the intelligence of the pre-trained language model. Based on this MoE architecture, we design an interleaved autoregressive-diffusion framework that generates text, audio, and behavior tokens, supervising the text and audio in an autoregressive manner and the behavior modality with diffusion loss. We further implement a diffusion forcing strategy to stabilize continuous generation. As a newly emerging and comprehensive task, we carefully design a humanoid agent evaluation protocol and achieve competitive performance in language understanding, audio-behavior alignment, and behavior execution for versatile multimodal humanoid generation.

## 1 INTRODUCTION

Recent advances in unified image-language understanding and generation, such as GPT-4o Hurst et al. (2024), Janus Wu et al. (2024), Transfusion Zhou et al. (2024), and Show-O Xie et al. (2024a), have demonstrated strong multi-task generalization and architectural scalability, which have driven the evolution of unified frameworks across language, vision, speech, and other modalities, such as human motion Jiang et al. (2023a) and robotic action Black et al.; Kim et al. (2024). Due to these unified multimodal foundations, humanoid generation, capable of mimicking human abilities in language understanding, speech generation, and behavior execution, should benefit robotics, human-computer interaction, filming, and virtual assistants.

Previous research on video-based human animation has explored diverse tasks, including portrait animation Guo et al. (2024); Tian et al. (2024; 2025), upper/full-body human animation Hu (2024); Hu et al. (2025a); Gan et al. (2025); Tan et al. (2024); Luo et al. (2025); Zhu et al. (2024); Men et al. (2024), and humanoid generation Ao (2024); Wang et al. (2025c). Recent portrait animation works Guo et al. (2024); Tian et al. (2024); Jiang et al. (2024) involve two conditions: face parameters Guo et al. (2024) or audio signals Tian et al. (2024); Jiang et al. (2024). The former, such as LivePortrait Guo et al. (2024), relies on intermediate representations like keypoints to efficiently drive portraits, while the latter, such as EMO Tian et al. (2024) and Loopy Jiang et al. (2024), favors an end-to-end approach to achieve audio-to-video generation without explicit facial modeling. Moreover, OmniTalker Wang et al. (2025c) proposes a cascaded approach that generates audio before talking-head animation, enabling finer-grained control. On the other hand, recent DiT-based human animation works, such as OmniHuman Lin et al. (2025a), EMO2 Tian et al. (2025), and

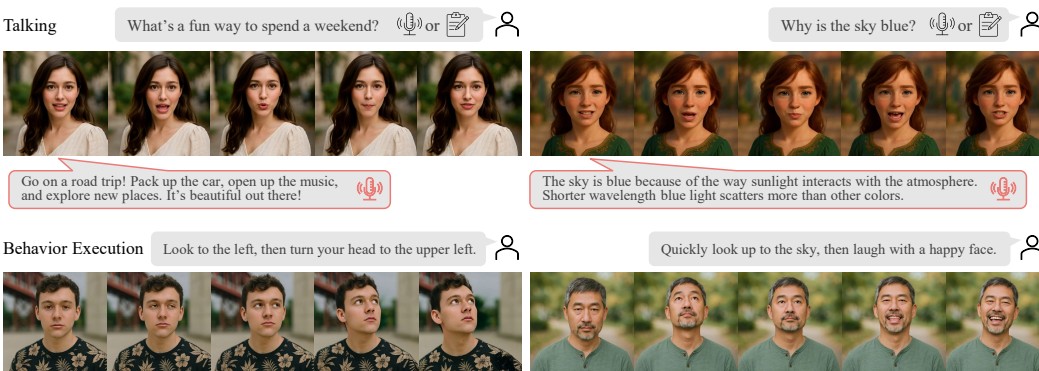

Figure 1: HumanExpert can address diverse humanoid generation tasks through various text or audio instructions. We provide the results on audio question-to-answer, talking head animation (the uppers), and behavior execution (the bottoms).

HunyuanCustom Hu et al. (2025b), benefit from general text-to-video backbones Yang et al. (2024); Kong et al. (2024); Seawead et al. (2025), achieving diverse conditional controls including text, audio, object interaction, and human identity, and producing more appearance-realistic and context-aware animations. However, most above methods primarily focus on animation control via text or audio cues, with generation processes that are typically modular or cascaded, often lacking unified modeling of human-like intelligence. Inspired by Body-of-Her Ao (2024) and other human-like generation works Zeng et al. (2024); Wang et al. (2025c); Xu et al. (2025), we thus aim to build a unified multimodal humanoid generative model that generalizes across language-audio understanding and speech-behavior generation, while mimicking human-like intelligence in both thoughts and behavior.

Two challenges are crucial for unified humanoid generations. The first is unifying the learning of continuous and discrete tokens within a single framework, and the second is solving the multimodal scaling problem without degrading the intelligence of pre-trained models. Insightfully, by decoupling model parameters across modalities Esser et al. (2024) and adopting a modality-expert training strategy Black et al.; Liang et al. (2024), a new expert can learn to generate new modality while preserving the original intelligence. Therefore, with recent advances in unified multimodal modeling, unified humanoid models equipped with a behavior expert can interleave continuous and discrete tokens in a single generative framework. Meanwhile, such frameworks preserve the capabilities of the pre-trained model, and support continued supervised fine-tuning on origin language modality for human-agent intelligence alignment.

In this work, we propose a unified multimodal humanoid generative model, HumanExpert, which leverages behavior expert to integrate strong large-scale audio-language models for humanoid tasks within a unified hybrid autoregressive-diffusion framework. To comprehend and generate human-like behaviors across different modalities, we first adopt a mixture-of-experts (MoE) architecture as a modality-independent backbone, wherein the behavior expert branch learns human behavior modeling interacting with language and audio latents, but avoid degrading the pre-trained intelligence. To efficiently train this MoE, we utilize an autoregressive-diffusion framework, supervising the text and audio with softmax loss and behavior with diffusion loss, generating text, audio, and behavior tokens in an interleaved manner. Moreover, we introduce a diffusion-forcing strategy to stabilize continuous behavior modeling, while history-based classifier-free guidance is employed to further improve output quality. We final incorporate DiT-based human video models to render behavior embeddings into realistic human videos. To evaluate our model, we carefully design a humanoid evaluation protocol and show that HumanExpert achieves competitive performance across language understanding (audio QA), audio-behavior alignment (lip sync), and behavior execution (listen-to-command).

We summarize our contributions as follows: (1) We propose a unified multimodal humanoid generative model, HumanExpert, which interleaves text, audio, and behavior tokens for human animation, and performs diverse humanoid tasks with one single model. (2) We introduce behavior experts within a mixture-of-experts backbone to comprehend with audio-language latents for efficient multimodal scaling, enabling the new humanoid learning without degrading the pre-trained intelligence. (3) We propose a general humanoid benchmark for multi-task humanoid evaluation, where HumanExpert shows competitive results across language understanding, speech generation, and behavior execution.

## 2 RELATED WORK

**Human Behaviour and Animation.** Ranging from 2D pose Cao et al. (2019); Andriluka et al. (2014); Khirodkar et al. (2024) to 3D motion Li et al. (2017); Oufqir et al. (2020); Romero et al. (2022); Loper et al. (2023); Pavlakos et al. (2019), and from explicit keypoints to implicit latents Wang et al. (2024d); Siarohin et al. (2019); Zhao & Zhang (2022); Zhao et al., **Human Behavior Representations** generally serve as a foundation for video-based human research, such as portrait animation Xie et al. (2024b); Tian et al. (2024); Guo et al. (2024); Jiang et al. (2024); Ji et al. (2024); Cui et al. (2024a); Zheng et al. (2024); Chen et al. (2025), upper/full-body human animation Hu (2024); Hu et al. (2025a); Gan et al. (2025); Lin et al. (2025a); Tian et al. (2025); Hu et al. (2025b), humanoid generation Ao (2024); Wang et al. (2025c), as well as 3D-based tasks including human motion generation Xin et al. (2023); Tevet et al. (2022); Jiang et al. (2023a); Zhang et al. (2023b); Petrovich et al. (2022), human-object interaction Pan et al. (2025a); Li et al. (2024a), and even humanoid robotics Cheng et al. (2024). Targeting videos, **Portrait Animation** synthesizes facial movements and expressions, often guided by 2D Zhou et al. (2020); Guo et al. (2024); Chen et al. (2019); Ji et al. (2021; 2022); Wei et al. (2024) and 3D face parameters Zhang et al. (2023a; 2021a); Kim et al. (2018); Zhang et al. (2021b); Yi et al. (2022); Zhang et al. (2023d), or directly by audio signals Jiang et al. (2024); Tian et al. (2024); Wang et al. (2024a;b); Cui et al. (2024a); Wang et al. (2025c). The former, such as MakeItTalk Zhou et al. (2020) and LivePortrait Guo et al. (2024), relies on intermediate representations like keypoints to efficiently drive portraits and achieve broad applicability, while the latter, such as EMO Tian et al. (2024) and Loopy Jiang et al. (2024), favors an end-to-end manner, by performing audio-to-video synthesis without explicit behavior modeling, aiming to reduce intermediate errors for robustness. In addition, OmniTalker Wang et al. (2025c) proposes a cascaded approach that generates audio before talking-head animation, enabling finer-grained control. Beyond faces, **Human Animation** Hu (2024); Tan et al. (2024); Luo et al. (2025); Men et al. (2024); Zhu et al. (2024) has advanced with DiT-based text-to-video models Yang et al. (2024); Kong et al. (2024); Seawead et al. (2025); Wan et al. (2025) and large pre-trained backbones, enabling flexible animation across face, upper body, and full body. Recent works, such as OmniHuman Lin et al. (2025a), HumanDiT Gan et al. (2025), EMO2 Tian et al. (2025), and HunyuanCustom Hu et al. (2025b), support diverse conditional controls including text, audio, object interaction, and character identity, achieving more appearance-realistic and context-aware animations. However, most human animations primarily focus on conditional control via audio or text cues, with generation processes typically modular or cascaded, often lacking unified multimodal understanding and generation. Inspired by Body-of-Her Ao (2024) and other human-like works Zeng et al. (2024); Wang et al. (2025c), **Humanoid Generation** offers a newly comprehensive task, we thus aim to synthesize humanoid agents that mimic deep human abilities such as language understanding, speech generation, and behavior execution within a unified framework.

**Unified Multimodal Understanding and Generation** has shown great potential in the image domain Wu et al. (2024); Xie et al. (2024a); Zhou et al. (2024); Pan et al. (2025b); Yu et al. (2024); Ma et al. (2024); Hurst et al. (2024); Team et al. (2023), often requiring complex and challenging architectures, ranging from hybrid encoding to hybrid backbones, to support unified vision-language understanding and generation. Meanwhile, these advances have also inspired broader tasks, such as vision-language modeling (VLM) Wang et al. (2024c); Lu et al. (2024); Liu et al. (2023; 2024), vision-language-action modeling (VLA) Black et al.; Kim et al. (2024), speech-language modeling Zeng et al. (2024); Team (2025); Xu et al. (2025), human motion modeling Jiang et al. (2023a), all evolving toward unified frameworks. While language models primarily rely on discrete tokens for symbolic modalities, perceptual modalities like vision and human behavior benefit more from continuous representations Wang et al. (2025b); Li et al. (2024b). To address this gap, **Hybrid Encoding** has been proposed to support both semantic-level and pixel-level understanding and generation, enabling flexible encoding strategies across and within modalities to improve cross-modal alignment. While TokenBridge Wang et al. (2025b) decouples discretization from tokenizer training by applying post-training quantization to extract discrete tokens directly from continuous tokens, more recent work Wu et al. (2024); Zhou et al. (2024); Xie et al. (2024a) adopts hybrid encoding schemes to ease the burden of multimodal representation learning. Relying on such encodings, **Hybrid Backbone** architectures have progressed from pure autoregressive models (using discrete tokens) or pure diffusion models (using continuous tokens) to unified frameworks that combine both objectives, such as Transfusion Zhou et al. (2024) and Show-O Xie et al. (2024a), which adaptively learn from mixed modalities. To further enhance training in these settings, Diffusion Forcing Chen et al. (2024) introduces a new paradigm where a diffusion model denoises tokens with independently sampled

noise levels under a next-token prediction framework. Building upon this, DFoT Song et al. (2025) incorporates history-guided strategies across temporal and frequency dimensions to improve video dynamics and generalization. Despite these advances in vision-language and multi-task generation, the development of unified multimodal models, capable of generating humanoid agents that mimic human behavior and intelligence, remains limited and underexplored.

**Mixture of Multimodal Experts**. With the advancement of pre-trained large models, full-parameter fine-tuning on dense models has shown potential for performance gains, but also poses economic inefficiencies and substantial scaling challenges in downstream tasks. To address these issues, **Sparse Models** have been widely explored, including LoRA Hu et al. (2022), adapters Zhang et al. (2023c); Poth et al. (2023), mixture-of-experts (MoE) Jacobs et al. (1991), modality-specific branches Esser et al. (2024), and mixture-of-transformers(MoT) Liang et al. (2024); Shi et al. (2024); Black et al., offering flexible solutions for multimodal and multi-stage training. Among them, LoRA Hu et al. (2022) is the most widely adopted due to its simplicity and effectiveness, especially in tasks such as image stylization Wang et al. (2023) and vision adaptation Wang et al. (2025a). However, its applicability to multimodal output generation remains limited. Recently, **Multimodal Expert** Esser et al. (2024); Black et al. has demonstrated strong learning capabilities. For instance, Stable Diffusion 3 Esser et al. (2024) proposes MM-DiT, which uses separate weights for the text-image modalities and enables bidirectional information flow, thereby improving text comprehension. Moreover, $\pi0$ Black et al. builds on a pre-trained VLM by incorporating a separate action expert to generate continuous actions via flow matching, enabling fine-grained control in embodied generation. Similarly, MoT Liang et al. (2024) proposes a sparse multimodal transformer architecture to reduce pretraining costs across text, image, and speech inputs, and LMFusion Shi et al. (2024) extends this adaptive strategy to unify image understanding and generation. Inspired by $\pi0$ Black et al. and MoT Liang et al. (2024), we propose *HumanExpert*, a framework for humanoid generation that decouples model parameters across modalities with behavior expert, which can reduce multimodal scaling costs, avoid downgrading pre-trained language models, and enable the interleaving of discrete and continuous tokens within a unified generative framework.

## 3 METHOD

We introduce HumanExpert, a unified generative framework that extends pretrained text-only LLMs Touvron et al. (2023); Xu et al. (2025) and text–speech LLMs Yao et al. (2024); Zeng et al. (2024) with behavior generation for natural embodied interaction. As shown in Fig. 2, HumanExpert builds on a transformer backbone and replaces non-embedding components—feed-forward blocks, attention, and layer norms—with modality-specific experts, following Black et al.; Liang et al. (2024); Shi et al. (2024). Modalities interact through shared cross-modal attention for unified understanding. Text and speech are produced autoregressively, while behavior is generated via holistic denoising with per-token noise levels using an expert part model. This hybrid design preserves causal semantics for language/audio and enables long-horizon, stable behavior synthesis without regenerating past tokens.

Formally, given text $w^{1:N}$, we tokenize into $L_t$ tokens using Tiktoken (vocabulary size $K_t$). Spoken input $s$ is discretized by a pretrained speech tokenizer $\mathcal{E}s$ into a sequence of $L$ speech tokens. Conditioned on text or speech, HumanExpert outputs (1) a sequence of discrete tokens $z^{1:L\text{out}}$ for text/speech, which detokenize to $w_{\text{out}}$ or decode to $s_{\text{out}}$, and (2) a behavior sequence $m_{\text{out}}$. We define behavior as disentangled motion in two forms: (i) head animation—a 1D identity-agnostic latent $m \in \mathbb{R}^{T\times512}$ at 25 fps learned jointly with a video generator; (ii) whole-body animation—normalized 2D pose keypoints from RTMPose Jiang et al. (2023b), where 133 keypoints $j \in \mathbb{R}^{T\times266}$ are normalized by the character center $c \in \mathbb{R}^{T\times2}$ and bounding-box size $b \in \mathbb{R}^{T\times1}$, producing $m \in \mathbb{R}^{T\times271}$ with values scaled to $[-1, 1]$. A DiT-based renderer converts $m_{\text{out}}$ into video $v_{\text{out}}$ with generated speech.

### 3.1 MODEL ARCHITECTURE

HumanExpert consists of two structurally similar but parametrically independent expert modules. The text-speech expert handles both text and speech modalities, generating discrete token sequences in an autoregressive manner. The behavior expert, on the other hand, focuses on the behavior modality, generating motion sequences via a token-level denoising diffusion process.

Figure 2: Architecture overview: HumanExpert consists of text-speech expert and behavior expert. By leveraging multimodal embeddings, behavior expert generates continuous behavior tokens by diffusion forcing, while text-speech expert generates discrete text or audio tokens by next-token prediction in a unified framework.

**Text-speech Expert** first employs a pretrained speech tokenizer from Zeng et al. (2024) to encode a speech segment $s$ of duration $T$ seconds to a sequence of discrete speech tokens $z^{1:L} = \{z^i\}_{i=1}^L$. These tokens are embedded in a shared vocabulary space, allowing seamless integration with language representations. The tokenizer comprises two components: an encoder $\mathcal{E}$ and a conditional flow matching model $\mathcal{D}$. The encoder $\mathcal{E}$ is based on Whisper Radford et al. (2023), a multilingual ASR model. It processes the input Mel spectrogram $s_{\text{mel}}$ through the Whisper subnetwork, then applies 1D average pooling to reduce the temporal resolution by a factor of $k$. The resulting features are discretized via a vector quantizer, producing semantic tokens $z^{1:L}$. Each token $z^i$ corresponds to the nearest codebook vector, with the codebook updated during training using exponential moving averages (EMA) and a random restart strategy Dhariwal et al. (2020). The flow matching model $\mathcal{D}$ reconstructs Mel spectrograms conditioned on discrete tokens $z^{1:L}$. It learns a transformation from a simple prior distribution $p_0(s_{\text{mel}})$ to the target distribution $q(s_{\text{mel}})$, using a time-dependent vector field trained via an optimal transport objective. Finally, a HiFi-GAN vocoder converts the generated spectrograms into speech waveforms. Together, $\mathcal{E}$ and $\mathcal{D}$ enable speech to be represented as discrete tokens that are semantically aligned with language. To jointly model text and speech, we adopt a decoder-only transformer initialized from a pretrained language model. The original language vocabulary $V_t = \{v_t^i\}_{i=1}^{K_t}$ is extended with a speech vocabulary $V_s = \{v_s^i\}_{i=1}^{K_s}$, which preserves the ordering of the speech tokenizer's codebook $Z$. This results in a unified vocabulary $V = \{V_t, V_s\}$, allowing both input and output sequences to include a mix of text and speech tokens. Using this unified token space, the model can model diverse speech-language tasks in a consistent format.

**Behavior Expert** adopts the architectural backbone of pretrained language models, but with behavior-specific sizes and parameters. While structurally similar, it operates independently of the text-speech expert. To enable joint reasoning across modalities, HumanExpert integrates behavior with text and speech through shared cross-modal attention layers, promoting unified multimodal understanding and generation. Specifically, we extend the decoder-only transformer with behavior-specific self-attention and feed-forward networks (FFNs), analogous to the image and text branches. Each behavior input $m$ is first projected into a sequence of latent embeddings $x_t^{\text{m}}$. These latent embeddings are processed using behavior-specific query, key, value (QKV) projections, and attention layers. To facilitate cross-modal interaction, we compute QKV representations by interleaving modality-specific hidden states. Specifically, we apply separate linear projections to the hidden states of the text-speech modalities $e^{(t,s)}$ and behavior $e^{(m)}$. The Q, K, and V at position $i$ are obtained by summing the modality-specific projections at the corresponding positions and applying a hybrid attention mask tailored for chunk-level streaming. Thus, text queries are strictly causal, while motion queries may attend bidirectionally within the current chunk and to any past chunks, but never to future chunks.

Formally, this is expressed as:

$$Q^{t,s,m} = e^{(t,s)}W_Q^{t,s} + e^{(m)}W_Q^m$$
$$K^{t,s,m} = e^{(t,s)}W_K^{t,s} + e^{(m)}W_K^m \qquad (1)$$
$$V^{t,s,m} = e^{(t,s)}W_V^{t,s} + e^{(m)}W_V^m$$

As illustrated in Fig. 2, cross-modal attention is then applied using a hybrid attention mask, where text and speech tokens attend only within their own modalities, while behavior tokens attend to all modalities:

$$h_O^{\mathrm{m}} = \mathrm{O_m}\left(\mathrm{softmax}\left(\frac{Q^{t,s,m}(K^{t,s,m})^T + \mathrm{Mask}^m}{\sqrt{d}}\right)V^{t,s,m}\right) \qquad (2)$$

The resulting hidden states are passed through behavior-specific FFNs, producing contextually enriched representations. Finally, these are projected into behavior noise predictions. This modular design allows the behavior expert to model both low-level motor dynamics and high-level semantic intent, while benefiting from shared cross-modal context through interleaved attention.

## 3.2 TRAINING OBJECTIVES

**Text-speech Expert** mixed text and speech tokens from a shared vocabulary $V$. The source sequence is denoted as $X_{\mathrm{in}} = \{x_{\mathrm{in}}^i\}_{i=1}^N$, and the target sequence as $X_{\mathrm{out}} = \{x_{\mathrm{out}}^i\}_{i=1}^L$, where $N$ and $L$ are the lengths of the input and output sequences, respectively. As illustrated in Fig. 2, the input tokens are passed through a transformer decoder that generates the target sequence in an autoregressive fashion. At each time step, the model predicts the probability of the next token conditioned on the previously generated tokens and the source input:

$$p_\theta(X_{\mathrm{out}} \mid X_{\mathrm{in}}) = \prod_{i=1}^L p_\theta(x_{\mathrm{out}}^i \mid x_{\mathrm{out}}^{<i}, X_{\mathrm{in}}) \qquad (3)$$

This part is trained to maximize the log-likelihood of the target sequence, using following objective:

$$\mathcal{L}_{\mathrm{LM}} = -\sum_{i=1}^L \log p_\theta(x_{\mathrm{out}}^i \mid x_{\mathrm{out}}^{<i}, X_{\mathrm{in}}) \qquad (4)$$

**Behavior Expert** operates on sequences of behavior tokens $\{x_m^t\}_{t=1}^T$, where each token may be subject to a different degree of partial noising. Unlike the text-speech expert, which is trained autoregressively, the behavior expert adopts a non-autoregressive training objective, enabling the model to handle varying noise levels across time steps. To model this, we denote the noisy behavior sequence as $\{x_m^{k_t}\}_{t=1}^T$, where $k_t \in [0, K]$ indicates the noise level of token $x_m^t$. The behavior expert consists of a recurrent dynamics function $p_\theta(z_t \mid z_{t-1}, x_{\mathrm{in}}, x_m^{k_t}, k_t)$ and an observation model $p_\theta(x_m^0 \mid z_t)$ that together learn to denoise the input sequence by minimizing the expected prediction error over varying noise levels. The overall training objective follows a noise-prediction paradigm:

$$\mathcal{L}_{\mathrm{DF}} = \mathop{\mathbb{E}}_{\substack{k_t, x_m, \epsilon_t \\ z_t \sim p_\theta(z_t \mid z_{t-1}, x_{\mathrm{in}}, x_m^{k_t}, k_t)}} \sum_{t=1}^T \left\| \epsilon_t - \epsilon_\theta(z_{t-1}, x_{\mathrm{in}}, x_m^{k_t}, k_t) \right\|^2 \qquad (5)$$

Here, $\epsilon_t$ is the added Gaussian noise at timestep $t$, and $\epsilon_\theta$ predicts the noise based on the previous latent state $z_{t-1}$ and the noisy token $x_m^{k_t}$. The behavior expert thus learns to reconstruct clean behavior tokens by leveraging both temporal context and the structure of noise, enabling robust behavior generation even under partial observability or corrupted inputs. Moreover, to enable classifier-free guidance at inference, we randomly drop text and audio conditions with probability $p$ for training. The overall training objective combines both components:

$$\mathcal{L}_{\mathrm{total}} = \lambda_{\mathrm{LM}} \cdot \mathcal{L}_{\mathrm{LM}} + \lambda_{\mathrm{DF}} \cdot \mathcal{L}_{\mathrm{DF}} \qquad (6)$$

## 3.3 INFERENCE

HumanExpert outputs each modality using distinct sampling strategies. The text-speech expert generates discrete tokens autoregressively, while the behavior expert synthesizes continuous motion sequences through a structured denoising process guided by a noise schedule matrix.

**Text-speech Expert** generates the target sequence token by token, recursively sampling from the predicted distribution:

$$p_\theta \left( \hat{x_t}^i \mid \hat{x_t}^{<i}, x_s \right). \tag{7}$$

This autoregressive decoding conditions each token on both the source input and the previously generated tokens, allowing it to produce coherent and contextually grounded text or speech outputs.

**Behavior Expert** follows the sampling procedure by denoising from a noise schedule matrix $\mathcal{K} \in [K]^{M \times T}$ defined over a 2D grid, where each row $m$ corresponds to a behavior sequence and each column $t$ to a time step. Each entry $\mathcal{K}_{m,t}$ specifies the desired noise level of token $x_m^t$ during generation. To synthesize a behavior sequence of length $T$, we initialize $\{x_m^t\}_{t=1}^T$ with Gaussian noise $\mathcal{N}(0, \mathbf{I})$, corresponding to $\mathcal{K}_{M,t} = K$. The module then denoises row-by-row from $m = M$ to $m = 0$, and left-to-right within each row, progressively reducing noise based on the prescribed schedule $\mathcal{K}$. At the final row $m = 0$, the denoised output $\{x_0^t\}_{t=1}^T$ constitutes the generated behavior sequence, where $\mathcal{K}_{0,t} = 0$ ensures that each token is fully reconstructed. This procedure enables the behavior expert to flexibly control the denoising rate for different tokens and time steps. Because the module is trained to handle arbitrary sequences of noise levels, $\mathcal{K}$ can be customized to adaptively govern behavior synthesis strategies, such as uniform denoising, left-to-right generation, or other user-specified temporal schedules, without retraining the model.

**Classifier-Free Guidance**(CFG). After the next-token prediction in the text-speech steam, the behavior expert functions as a conditional diffusion model that incorporates all text/audio hidden states, often called key-value (KV) caches, and behavior history. Therefore, we leverage these two types of conditions to apply CFG, which enables more accurate control and improves sample quality. Specifically, we adopt history guidance Song et al. (2025) during sampling, employing three conditioning strategies: 1) full history, including text/audio context and behavior history; 2) partial condition, with text/audio context and low-frequency behavior history; and 3) fully masked condition. We also implement a vanilla history guidance that uses only two strategies: full-history context and fully masked context, providing a balance between performance and runtime efficiency. In addition, we follow the pyramid scheduling strategy from Diffusion Forcing Chen et al. (2024), where tokens in the far future are assigned higher noise levels than those in the near future. This schedule offers a favorable trade-off between effectiveness and computational cost.

**Humanoid Agents Interfaces.** To produce visually expressive humanoid agents, the generated behavior sequence $m_{\text{out}}$ is transformed into photorealistic video frames using a hybrid rendering architecture Zhao et al. or DiT-based human animation backbone Lin et al. (2025a). These renderers integrate pretrained diffusion models $\mathcal{G}_{\text{D}}$ with a reference-conditioned network $\mathcal{G}_{\text{R}}$, allowing for both motion fidelity and visual consistency. The behavior sequence $m_{\text{out}}$ is injected into the denoising backbone of $\mathcal{G}_{\text{D}}$ via motion-guided cross-attention layers, which modulate temporal dynamics while preserving appearance based on a given reference image $I_R$ (e.g., a portrait or full-body photo). To further ensure identity consistency and suppress visual artifacts from the driving motion, the reference encoder $\mathcal{G}_{\text{R}}$ extracts fine-grained appearance features $f_{\text{app}}$ from $I_R$. These features are fused with intermediate motion representations in the generator, enabling visual rendering of humanoid agents.

# 4 EXPERIMENTS

Extensive comparisons evaluate the performance of our HumanExpert across multiple motion-relevant tasks and datasets. We evaluate HumanExpert across multiple behavior-focused tasks and datasets. Datasets, metrics, and implementation details are provided in Sec. 4.1. We first evaluate multimodal generation by comparing against state-of-the-art audio-driven portrait animation methods in Sec. 4.2. As these methods are limited in behavior execution, we further compare with identity-consistent, text-conditioned video generation baselines. To validate our expert design, we compare against full-parameter fine-tuning and low-rank adaptation (LoRA). Results in Sec. 4.3 show that our approach preserves language capabilities while effectively extending to new modalities. Additional qualitative results, extended ablations, and implementation details are provided in the supplementary materials.

## 4.1 EXPERIMENTAL SETUP

**Datasets.** To support both text-to-behavior generation and behavioral command execution, we curate a hybrid dataset comprising 36k text–speech–video triplets and an additional 3k videos paired with

Table 1: Comparison of behavior execution. HumanExpert outperforms text-conditioned video baselines on silent behavior execution tasks, achieving better motion accuracy and visual coherence.

| Method | SSIM ↑ | LPIPS ↓ | FVD ↓ | ID-SIM ↑ | EMO-SIM ↑ |
|---|---|---|---|---|---|
| ConsisID (Yuan et al. (2025)) | 0.414 | 0.604 | 2487 | 0.379 | **0.229** |
| Phantom (Liu et al. (2025b)) | 0.367 | 0.582 | 1899 | 0.647 | -0.004 |
| **HumanExpert** | **0.538** | **0.306** | **792.6** | **0.723** | 0.137 |

Table 2: Comparison of audio–behavior alignment on CelebV-HQ. HumanExpert achieves competitive performance and uniquely supports text-to-audio and silent-behavior generation—capabilities not available in baseline methods.

| Method | SSIM ↑ | LPIPS ↓ | FID ↓ | FVD | Sync-C ↑ | Sync-D ↓ |
|---|---|---|---|---|---|---|
| JoyVasa (Cao et al. (2024)) | **0.605** | **0.046** | 79.3 | 600.3 | 3.429 | 8.667 |
| EchoMimic (Chen et al. (2025)) | 0.486 | 0.400 | 113.4 | 815.4 | 2.331 | 9.574 |
| Memo (Zheng et al. (2024)) | 0.597 | 0.054 | **77.3** | **536.2** | 3.426 | 8.270 |
| Hallo3 (Cui et al. (2024b)) | 0.564 | 0.121 | 86.0 | 601.2 | 3.242 | 9.315 |
| Sonic (Ji et al. (2024)) | 0.491 | 0.168 | 108.5 | 585.0 | 3.650 | **6.784** |
| **HumanExpert** | 0.556 | 0.237 | 91.3 | 636.1 | **3.743** | 8.935 |

command–response annotations. The triplets are sourced from publicly available datasets Zhang et al. (2021c); Xie et al. (2022); Kirschstein et al. (2023). The command set was collected with adult participants who provided written consent and received fair compensation. We evaluate on three curated sets. (1) A celebrity video benchmark of 100 videos sampled from the CelebV-HQ dataset Zhu et al. (2022).(2) A behavioral-commands benchmark of 100 test videos depicting varied command-execution behaviors. (3) A portrait benchmark of 100 in-the-wild reference images drawn from DeviantArt DeviantArt (2025), GPT-4o Hurst et al. (2024), and Pexels Pexels (2025), covering diverse facial structures, appearances, and visual styles. Together, these sets span wide appearance and motion diversity and support consistent, cross-domain evaluation. Further details and qualitative examples are provided in the supplementary materials.

**Evaluation Metrics** are summarized as two main aspects: (1) Image and Video Quality. To assess visual fidelity, we compute both structural and perceptual image-level metrics, including Structural Similarity Index (SSIM) and Learned Perceptual Image Patch Similarity (LPIPS). For temporal consistency and overall video quality, we adopt the Fréchet Video Distance (FVD) Skorokhodov et al. (2022). (2) Behavioral Alignment. To evaluate identity preservation, we use ArcFace Deng et al. (2019) to compute cosine similarity between identity embeddings (ID-SIM). For emotion consistency, we leverage a pretrained emotion recognition model, EmoNet Toisoul et al. (2021), and report the mean of Concordance Correlation Coefficient (CCC) and Pearson Correlation Coefficient across both valence and arousal dimensions (EMO-SIM). We also evaluate audio-visual synchronization with Sync-C and Sync-D Chung & Zisserman (2016).

**Implementation Details.** We adopt GLM-4-Voice Zeng et al. (2024) as the backbone architecture for our speech-language expert, using a 40-layer transformer as the base model. The feed-forward networks are configured with dimensionalities of $d_{\text{ff}}^{t,s} = 13{,}696$ for the text-speech expert and $d_{\text{ff}}^{m} = 6{,}848$ for the behavior expert. Attention layers use an inner dimensionality of $d_{\text{kv}} = 128$, while hidden representations have dimensionalities of $d^{t,s} = 4{,}096$ and $d^{m} = 2{,}048$ for the two experts, respectively. The text-speech expert loss $\lambda_{\text{LM}}$ is set to zero, and all text-speech modules are kept frozen during training. For the diffusion-based behavior generation, we apply classifier-free guidance with a dropout probability of $p = 0.1$. During inference, we use a DDIM sampler with 50 denoising steps and set the guidance scale to $w = 1.5$. All models are optimized using the AdamW optimizer. The learning rate is set to $1 \times 10^{-4}$ during the audio-behavior alignment stage and $2 \times 10^{-5}$ during full fine-tuning. We train using a mini-batch size of 4 for both stages. The language model is trained for 300K iterations during the alignment phase, followed by an additional 300K iterations to incorporate the command execution task.

## 4.2 QUANTITATIVE RESULTS

**Comparisons on Behavior Execution.** Some actions—such as blinking, head turns, or micro-expressions—occur without sound and cannot be handled by audio-driven methods. In contrast,

Table 3: Ablation of Behavior Expert. Expert separation in HumanExpert preserves reasoning abilities. 'S→T': speech-to-text; 'T': text-to-text.

| Method | Web Questions | | Llama Questions | | TriviaQA | |
|---|---|---|---|---|---|---|
| | T | S → T | T | S → T | T | S → T |
| w/ FT | 1.15 | 1.13 | 1.26 | 1.17 | 1.03 | 1.01 |
| w/ LoRA | 6.60 | 6.34 | 7.63 | 7.08 | 5.41 | 5.26 |
| **HumanExpert** | **6.69** | **6.40** | **7.65** | **7.13** | **5.43** | **5.32** |

HumanExpert supports such behaviors through unified modeling of text, speech, and behavior. Since no existing methods address this setting, we compare against text-conditioned video generation models with identity consistency. As shown in Tab. 3, HumanExpert achieves clearly superior results in both motion accuracy and visual coherence.

**Comparisons on Audio-Based Portrait Animation.** This task involves generating talking head videos conditioned on an audio input. While HumanExpert is capable of producing responses from either text or audio prompts, for fair comparison, we apply teacher forcing to align the output of the text-speech expert with the given driving audio. We compare our method with recent state-of-the-art approaches Zheng et al. (2024); Cao et al. (2024); Chen et al. (2025); Ji et al. (2024); Cui et al. (2024b), focusing on motion quality and identity consistency. As shown in Tab. 2, HumanExpert achieves competitive results across most metrics, despite offering broader capabilities such as audio generation from text prompts and silent behavior execution, which are beyond the scope of baseline methods.

### 4.3 ABLATION STUDIES

HumanExpert is designed with two structurally similar but parametrically independent expert modules to preserve the language model's original reasoning capabilities. We conduct ablations to validate this design choice. Additional results are included in the supplementary materials.

**Architecture.** We compare two model variants: (1) Full Fine-tuning: All language model parameters are updated, with added behavior-specific embeddings and output heads. While this enables behavior generation, it significantly impairs language reasoning due to the imbalance in training data, which emphasizes alignment over QA. (2) LoRA-based Adaptation: The language model is frozen, and behavior capabilities are introduced via low-rank adaptation. This preserves language understanding but limits scalability due to fewer trainable parameters. These results highlight the value of modular expert separation in maintaining strong language performance while extending to new modalities.

## 5 DISSCUSION

As a new trial to explore humanoid generation through unified multimodal models, the proposed HumanExpert still has the following limitations. HumanExpert utilizes human behavior to represent facial expressions and articulated bodies, whereas other existing works focus on video-based humanoid agents Ao (2024); Tian et al. (2025). Besides, our method is also restricted to multiple humans without modeling human-object Liu et al. (2025a), or human-environment interactions Ao (2024). It is promising to extend our framework to real-time or instant video generation settings Lin et al. (2025b); Frans et al. (2024), and to jointly generate controllable humans and interactive environments within a unified world model.

We summarize HumanExpert as a unified text-audio-behavior framework that enables humanoid agent tasks via behavior experts. Compared to prior works in human animation Cui et al. (2024b); Ji et al. (2024), HumanExpert achieves competitive results in portrait animation, behavior execution, and audio-based QA within a single framework. With the progress of multimodal models Liang et al. (2024); Black et al., HumanExpert is capable of generating interleaved text-audio-behavior sequences and shows competitive performance in language understanding, audio-behavior alignment, and behavior execution, supporting its effectiveness for general multimodal humanoid generation.

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
