# OpenReview forum: "HumanExpert: Unified Multimodal Humanoid Generation"
_ICLR.cc/2026/Conference — ICLR 2026 Conference Withdrawn Submission_

### Official Review · Reviewer_drme · 2025-10-24

**Soundness:** 3
**Presentation:** 2
**Contribution:** 2
**Rating:** 4
**Confidence:** 4

**Summary:**

This paper introduces HumanExpert, a hybrid generative framework that combines autoregressive text/speech modeling with behavior diffusion experts to achieve human-like multimodal generation. The model is designed to understand language and speech while generating synchronized speech and bodily motion. A key architectural choice is the use of a mixture-of-experts (MoE) scheme, where a dedicated "behavior expert" is integrated to avoid diluting the capabilities of the pre-trained language model. The behavioral component is generated via Diffusion Forcing and historically-guided classifier-free guidance (CFG), and later rendered into video using a DiT-style renderer that translates behavior embeddings into visual sequences. The authors also propose a multitask evaluation protocol and report competitive results on speech-avatar synchronization, silent motion execution, and other multimodal tasks.

**Strengths:**

1.	The framework unifies discrete (text/speech) and continuous (action) modalities in a coherent generative setup. The division of labor between autoregressive modeling for language and diffusion modeling for actions is intuitively reasonable and technically well-executed.
2.	The decoupling of multimodal parameters via a dedicated "behavior expert" is theoretically sound. It minimizes interference with the language model's pre-trained capacity, and the paper clearly details the associated attention/masking strategies—indicating strong extensibility.
3.	A custom evaluation protocol is presented to assess multiple capabilities within a single model, including speech-driven talking avatars and silent-action command execution.

**Weaknesses:**

W1. Inconsistent narrative around "unified generation" and "single-model" claims. While the paper claims to achieve single-model unified generation, the actual process involves generating behavior embeddings which are then passed to an external DiT/hybrid renderer for final video synthesis. This two-stage setup seems to conflict with the narrative of full end-to-end unification. The authors should clarify how behavior expert outputs couple with the renderer—are these modules swappable? What is the renderer's standalone contribution to final performance?

W2. Claims of preserving linguistic intelligence contradict training details. The paper emphasizes preserving the capabilities of the original language model, yet the implementation freezes the text-speech experts (λ_LM = 0), meaning the language pathway isn’t jointly optimized. If no fine-tuning is performed on language, then the claim of preserving intelligence may stem more from non-interference than architectural merit. A controlled comparison against FT/LoRA baselines using standardized language tasks would help clarify this.

W3. Unidirectional cross-modal attention may hinder full integration. The masking strategy enforces that text/speech tokens cannot attend to behavior tokens, while the reverse is allowed. This asymmetry prevents behavior history from influencing future language generation—a potential bottleneck for closed-loop tasks such as dialogue grounding. The rationale behind this design choice and comparative results with bidirectional attention should be discussed.

W4. Limited data and insufficient statistical rigor. The training corpus consists of only 36k triplets and 3k videos; each benchmark test set includes merely ~100 samples (CelebV-HQ, behavioral commands, portrait). This low scale raises concerns about generalizability. Confidence intervals, bootstrap resampling, or paired tests (e.g., McNemar or permutation tests) are necessary to assess reliability.

W5. Misalignment between metrics and task fidelity. Metrics such as SSIM, LPIPS, FVD, and identity/emotion similarity are used to evaluate action execution, but these fail to capture instruction-level correctness (e.g., keypoint trajectory deviations or alignment of motion phases). Similarly, audio-visual synchronization is only assessed in talking-head scenarios. Structured evaluation for text/speech-to-motion alignment is needed.

W6. Weak comparison to state-of-the-art multimodal baselines. Tables 1 and 2 mainly compare with unimodal or task-specific baselines. There's a lack of direct benchmarking against recent unified multimodal models (e.g., VLA, π0, audio-language LLMs). Even if functionality differs, comparable protocols and shared metrics should be established.

W7. Results are competitive but not consistently superior. In Table 2, the model does not outperform baselines across all LPIPS/FID/FVD metrics. Claims of competitiveness should be supported with statistical significance testing and error bars, alongside analyses explaining performance gaps (e.g., rendering pipeline, prior knowledge injection, dataset size).

W8. Limited ablation on key mechanisms. There’s no thorough breakdown of the individual contributions from Diffusion Forcing, historical CFG, or the noise schedule matrix K. Claims that "expert separation" preserves linguistic capability better than FT/LoRA are based on sparse evidence. Clarify the units in Table 3 and what the reported values signify.

W9. Reproducibility and compute transparency are lacking. While optimizer, learning rate, steps, batch size, and DDIM steps are reported, key details such as total compute resources, rendering costs, training duration, and open-sourcing plans are omitted.

W10. Ethical considerations are underdeveloped. The system leverages identifiable data (e.g., celebrity faces, synthetic voices), raising concerns around deepfakes, portrait rights, copyright, and data licensing. The paper briefly mentions participant consent and compensation but lacks a detailed response to ICLR’s ethics checklist. Legal status of datasets and models (CelebV-HQ, DeviantArt, GPT-4o, Pexels, GLM-4-Voice) must be clarified.

W11. Writing and terminology inconsistencies. There are spelling/formatting errors (e.g., “DISSCUSION”) and inconsistent use of terms (e.g., behavior tokens vs. embeddings vs. latents). A unified terminology section and abbreviation table would improve readability.

**Questions:**

1.	Why were the text/speech modules completely frozen (λ_LM = 0)? Would enabling joint SFT affect the conclusion about preserving linguistic competence? Please provide catastrophic forgetting comparisons between FT, LoRA, and your approach.

2.	Is the unidirectional attention (text/speech not seeing behavior) a stability or efficiency choice? What changes when bidirectional attention is enabled?

3.	How will you structurally evaluate action correctness? Can you provide quantitative indicators such as keypoint alignment, angular velocity, or temporal lag with significance tests?

4.	What's the individual impact of Diffusion Forcing, CFG, and the noise schedule matrix K? How do DDIM steps and guidance scale influence the speed-quality tradeoff?

5.	Can the renderer be swapped? What is its marginal contribution to final performance? Any biases introduced by renderer dependencies?

6.	Are the datasets (CelebV-HQ, DeviantArt, Pexels, GPT-4o) licensed and publicly accessible? Will controlled access or statistical summaries be offered for reproducibility?

Suggestions for Improvement

1.	Report 95% CIs and paired statistical tests for all primary and ablation results; expand test sets from 100 to at least 500 samples, or stratify by subject ID.

2.	Include protocolized comparison against strong unified multimodal baselines (even partial overlaps in capabilities warrant comparison).

3.	Provide traces of the contribution of key modules such as attention to "command tokens/speech segments/behavioural segments"; run counterfactual or permutation experiments to assess causality across modalities.

4.	Isolate the behavior predictor’s impact by swapping renderers and varying identity priors.

5.	Explicitly address all ICLR ethics checklist items (privacy, fairness, misuse, publication scope); outline usage restrictions, watermarking/detection plans, and accountability strategies.

---

### Official Review · Reviewer_3WiM · 2025-11-01

**Soundness:** 3
**Presentation:** 2
**Contribution:** 3
**Rating:** 6
**Confidence:** 3

**Summary:**

The paper proposes HumanExpert, a unified multimodal humanoid generative model that can take text or speech as input and generate text, audio, and human behavior (motion) in a single framework. The core observation is that language/speech are naturally discrete/autoregressive, while human behavior (head motion, body pose) is better modeled as continuous/diffusion. Trying to force everything into one modality usually either hurts the original LLM or makes behavior weak. So the authors decouple parameters per modality using a mixture-of-experts MoE-style backbone: a text–speech expert and a behavior expert trained to do motion with diffusion, but still able to attend to language/audio via shared cross-modal attention.

**Strengths:**

1. They identify a problem in “unified multimodal” framework: discrete (LLM, speech tokens) vs continuous (motion, human behavior).
2. Shared transformer shape but *parametrically independent experts for text/speech vs behavior matches works like SD3/MM-DiT/π0-style models.
3. On behavior execution they beat identity-consistent text-to-video models by a big margin.

**Weaknesses:**

1. “36k text–speech–video triplets + 3k videos + 100-size test subsets” is tiny for a paper claiming unified multimodal humanoid generation.
2. “The text-speech expert loss λLM is set to zero, and all text-speech modules are kept frozen.” That means the “unified” part is mostly one-way: behavior learns to use the LLM features, but the LLM does not learn to talk about behavior. Can the authors clarify this part?
3. The visual part is coming from an external renderer. That makes the core contribution more representation / conditioning than end-to-end generation. But I think this is nowhere near a major weakness.

**Questions:**

Why did you choose to make the behavior expert fully parameter-separated instead of using a lighter adapter/LoRA branch plus cross-attention? What failed when you tried the lighter variants?

How does inference latency compare to a pure-autoregressive baseline or to Transfusion-style models?

---

### Official Review · Reviewer_aJdK · 2025-11-01

**Soundness:** 2
**Presentation:** 2
**Contribution:** 2
**Rating:** 4
**Confidence:** 2

**Summary:**

HumanExpert presents a unified multimodal generative model for humanoid agents, integrating text, speech, and behavior through an autoregressive-diffusion framework with a mixture-of-experts (MoE) design. It achieves promising results on behavior execution and audio-visual alignment.

**Strengths:**

Tackles an ambitious and emerging task: general-purpose humanoid generation.

Well-designed modular architecture using modality-specific experts.

Combines discrete and continuous modalities in a hybrid framework.

**Weaknesses:**

- Claims support for speech ↔ behavior ↔ language interleaving, but most evaluations are unimodal (e.g., audio-driven portrait, silent behavior).

- Key ablations missing, e.g., impact of guidance, shared attention, or autoregressive vs. diffusion for behavior.

- Evaluation lacks depth:

Benchmarks are mostly curated by authors — potential bias and lack of reproducibility.

No human evaluation despite the subjective nature of motion quality and synchrony.

No efficiency or inference speed analysis, which is critical for real-time humanoid agents.


- Scalability and generalization unclear:

How does the model perform with unseen speakers, behaviors, or instructions?

No evidence on robustness to noise, domain shift, or ambiguous inputs.

**Questions:**

How does cross-modal attention avoid conflict or overfitting?

Can the model handle multi-modal conditioning (e.g., text → speech + behavior)?

What’s the inference latency for real-time use?

---

### Official Review · Reviewer_FVdR · 2025-11-01

**Soundness:** 2
**Presentation:** 2
**Contribution:** 3
**Rating:** 4
**Confidence:** 4

**Summary:**

This paper proposes HumanExpert, a unified multimodal model aiming to fuse language (discrete), speech (discrete), and behavior (continuous) modalities. Its core contribution is the use of a Mixture-of-Experts (MoE) architecture to add a "Behavior Expert" without harming the reasoning abilities of the pre-trained audio-language base model (GLM-4-Voice). The framework uses a novel "autoregressive-diffusion" hybrid mechanism for training: autoregression for text/speech and diffusion for behavior.

**Strengths:**

Frontier Problem: Addresses the key challenge of moving from "content generation" to "dynamic agent generation."

Novel Architecture: The "autoregressive-diffusion" hybrid framework is cleverly designed to handle discrete and continuous modalities uniformly.

Preserves Intelligence: The MoE design is highly effective. The ablation study (Table 3) proves it successfully avoids "catastrophic forgetting" of the base model's language capabilities when adding the new behavior modality.

**Weaknesses:**

Architectural Obscurity: There is a significant mismatch between the text and Figure 2. Key mechanisms described (e.g., cross-modal interaction, hybrid attention masks, chunks) are not visualized. Conversely, elements shown in the figure (like "Rearrange") are not explained, making the core architecture difficult to follow.

Vague Facial Representation: The paper is unclear on how facial "behavior" is defined and learned. It is described as an opaque "1D identity-agnostic latent," which is less interpretable than the explicit 2D keypoints used for body motion.

Based on the demo, it's difficult to judge whether the advantages in the results come from the motion model itself or from the renderer. I also found that Sonic's results look better. Since this model is for motion generation, it doesn't seem like a fair comparison against end-to-end video generation models."

**Questions:**

While the task is compelling and the method seems reasonable, the paper is generally unclearly written. This is compounded by a lack of discussion on how the approach fundamentally differs from other recent (though unreleased) works like "Body of Her" and "Midas" that also target this unified humanoid task.

MIDAS: Multimodal Interactive Digital-humAn Synthesis via Real-time Autoregressive Video Generation
Body of Her: A Preliminary Study on End-to-End Humanoid Agent

---

### Note · Authors · 2025-11-12

**Comment:**

Thank you for the helpful reviews. We are withdrawing the manuscript to incorporate the suggestions and will resubmit a revised version in the future.

**Withdrawal Confirmation:**

I have read and agree with the venue's withdrawal policy on behalf of myself and my co-authors.